# Two-dimensional lead halide perovskite lateral homojunctions enabled by phase pinning

Huilong Hong [1,4], Songhao Guo[2,4], Leyang Jin[1], Yuhong Mao[2], Yuguang Chen[1], Jiazhen Gu[1], Shaochuang Chen[1], Xu Huang[1], Yan Guan[1], Xiaotong Li[3], Yan Li [1], Xujie Lü [2] ✉ & Yongping Fu [1] ✉

Two-dimensional organic-inorganic hybrid halide perovskites possess diverse structural polymorphs with versatile physical properties, which can be controlled by order-disorder transition of the spacer cation, making them attractive for constructing semiconductor homojunctions. Here, we demonstrate a space-cation-dopant-induced phase stabilization approach to creating a lateral homojunction composed of ordered and disordered phases within a two-dimensional perovskite. By doping a small quantity of pentylammonium into (butylammonium)$_2$PbI$_4$ or vice versa, we effectively suppress the ordering transition of the spacer cation and the associated out-of-plane octahedral tilting in the inorganic framework, resulting in phase pining of the disordered phase when decreasing temperature or increasing pressure. This enables epitaxial growth of a two-dimensional perovskite homojunction with tunable optical properties under temperature and pressure stimuli, as well as directional exciton diffusion across the interface. Our results demonstrate a previously unexplored strategy for constructing two-dimensional perovskite heterostructures by thermodynamic tuning and spacer cation doping.

Semiconductor homojunctions, comprising the same material with distinct structural phases or dopants, show great promises in designing functional devices[1–3]. Two-dimensional (2D) lead halide perovskites (LHPs), as one of the most promising 2D semiconductors for optoelectronics, spintronics, and ferroelectrics, are highly attractive for constructing homojunctions due to their highly tunable compositions and structural phases that give rise to versatile physical properties[4–7]. The simplest 2D LHPs are quantum-wells superlattices consisting of PbX$_4^{2-}$ (X is a halide anion) layers of corner-sharing lead halide octahedra, alternating with spacer layers made of large organic ammonium cations (LA), resulting in a general formula of (LA)$_2$PbX$_4$. These structures exhibit a range of temperature- and pressure-dependent structural phase transitions, which involve the changes in conformational

and dynamic disorder of the LA cation and structural distortion of the PbX$_6^{4-}$ octahedra[8–11]. Consequently, different polymorphs of 2D perovskites exhibit varying optoelectronic properties such as bandgaps, exciton binding energies, work functions, and spin-orbit coupling strength[5,12], which offer exciting opportunities to create 2D homojunctions. Compared to previously demonstrated heterojunctions made of 3D LHPs and 2D LHPs with different halides[13–18], ion interdiffusion, which usually leads to unstable interfaces or gradient compositions in the heterojunctions[14,18–21], is expected to be largely suppressed in the homojunctions made of different polymorphs. It is worth noting that phase-engineered 2D homojunctions composed of concomitant polymorphs have been demonstrated with more traditional 2D materials, such as 2H-MoS$_2$ (trigonal prismatic) and 1T-MoS$_2$

[1]Beijing National Laboratory for Molecular Science, College of Chemistry and Molecular Engineering, Peking University, Beijing 100871, China. [2]Center for High Pressure Science and Technology Advanced Research, Shanghai 201203, China. [3]Department of Chemistry & Organic and Carbon Electronics Laboratories, North Carolina State University, Raleigh, NC 27695, USA. [4]These authors contributed equally: Huilong Hong, Songhao Guo. ✉e-mail: xujie.lu@hpstar.ac.cn; yfu@pku.edu.cn

(octahedral) or 1T′-MoS$_2$ (distorted octahedral), resulting in fascinating electronic, optical, and transport properties[1,22].

Temperature and pressure, as two of the most important thermodynamic variables, provide efficient methods for obtaining polymorphs of 2D LHPs, particularly through tuning the order-disorder transitions of the spacer cations which significantly influence inter-octahedral tilting and/or intra-octahedral distortion of the inorganic layer[23]. Dramatic changes in the optical and electronic properties induced by temperature or pressure have been observed in various 2D LHPs[9,24–31]. However, most of the 2D LHP polymorphs obtained under high pressure and high/low temperature are unstable at ambient condition, which limits the construction of homojunctions. Here, we demonstrate a spacer-cation-dopant-induced phase stabilization of the high-temperature polymorph under low-temperature conditions and the low-pressure polymorph under high-pressure conditions in 2D LHPs. Specifically, we find that by doping a small quantity of butylammonium cation (BA = $n$-C$_4$H$_9$NH$_3^+$) in (PA)$_2$PbI$_4$ (PA = $n$-C$_5$H$_{11}$NH$_3^+$) or vice versa, the ordering transition of the spacer cation can be suppressed, leading to the phase stabilization of the polymorph with disordered spacer cation when decreasing temperature or increasing pressure. Compared to the ordered phase, the dopant-stabilized disordered phase exhibits distinct responses in the structural and optical properties under temperature or pressure tuning. Built upon these discoveries, we develop homojunctions made of disordered and ordered phases of 2D LHPs via lateral epitaxial growth by introducing spacer cation dopants during the crystal growth of (PA)$_2$PbI$_4$ or (BA)$_2$PbI$_4$. We show effective tuning of the optical properties of the homojunctions under pressure and temperature stimuli, as well as directional exciton diffusion across the interface, which is driven by the built-in type-I band alignment.

## Results and discussion
### Doping-induced phase pinning

The structural phase behaviors of (BA)$_2$PbI$_4$ with temperature and pressure are shown in Fig. 1a, b[10]. As temperature decreases, (BA)$_2$PbI$_4$ exhibits a phase transition from a high-temperature disordered phase (*Pbca*) to a low-temperature ordered phase (*Pbca*) at ~274 K, which is driven by the ordering transition of the spacer cation. As the BA cations become ordered, the inorganic structure is locked in a larger out-of-plane octahedral tilting. The order-disorder phase transition can also be found at 0.2 GPa upon compression at room temperature (RT), which is identical to the temperature-induced phase transition[9]. As shown in Fig. 1b, initial compression and cooling have a similar effect that induces the ordering transition of the spacer cation. Intriguingly, we found this transition can be suppressed by doping with a small amount of PA cation (Fig. 1c). A series of (BA$_{1-x}$PA$_x$)$_2$PbI$_4$ single crystals

with varying doping ratios were synthesized (Methods). We found these doping structures crystallize in the high-temperature disordered phase of (BA)$_2$PbI$_4$ under ambient conditions, and they exhibit resistance to undergoing the disorder-order phase transitions when decreasing temperature or increasing pressure.

We first present the dopant-induced pinning of the disordered phase under low temperatures and its influence on optical properties. Single-crystal XRD (Supplementary Table 1) shows that the doping structures retain the high-temperature disordered phase at 80 K. The absence of phase transition in the doping structures has also been confirmed by differential scanning calorimetry analysis (Supplementary Fig. 1). To better understand the origin of phase retention, we compared temperature-dependent Raman spectra between (BA)$_2$PbI$_4$ and (BA$_{0.73}$PA$_{0.27}$)$_2$PbI$_4$. The low-frequency vibrational modes are mainly associated with the motions of the inorganic cages, whereas the high-frequency vibrational modes involve mainly the organic cations[28,32]. When increasing temperature from 77 to 363 K, (BA)$_2$PbI$_4$ shows an abrupt transition around 273 K, as evidenced by the jumps in the peak position and broadening of the low-frequency Raman modes (Fig. 2a). The peak broadening as well as redshift of these modes can be attributed to gradual activation of the octahedral tilting with increasing temperature[32]. The Raman spectra at high temperatures exhibits a broad central peak near zero frequency, which is typically observed in liquids, indicating dynamic disorder[33]. The high-frequency vibrational modes exhibit an abrupt change as well across the phase transition, which is associated with the disordering transition of the BA cation (Supplementary Fig. 2). By comparison, the Raman modes for both inorganic cages (Fig. 2b) and organic cations (Supplementary Fig. 2) in the (BA$_{0.73}$PA$_{0.27}$)$_2$PbI$_4$ continuously broaden with increasing temperature, showing the absence of an abrupt change.

By comparing the Raman spectra of the two structures, it is found that (BA$_{0.73}$PA$_{0.27}$)$_2$PbI$_4$ has a pronounced mode at 35 cm$^{-1}$ at low temperatures, which is absent in the ordered phase (BA)$_2$PbI$_4$. However, at room temperature, the Raman spectra of (BA$_{0.73}$PA$_{0.27}$)$_2$PbI$_4$ closely resemble the disordered phase of (BA)$_2$PbI$_4$, as the two structures have the same orthorhombic phase. By tracking the temperature-dependent evolution, we noted a pronounced redshift of this mode in (BA$_{0.73}$PA$_{0.27}$)$_2$PbI$_4$, shifting from 35 cm$^{-1}$ to 28 cm$^{-1}$ as temperature increased from 80 K to 300 K. This shift is accompanied by a significant broadening of the peak. Due to the significant redshift and broadening, the mode blends into an adjacent mode at 21 cm$^{-1}$, thus appearing as a peak shoulder at high temperatures (marked by * in Fig. 2a, b). In addition, this mode redshifts at a rate of 0.04 cm$^{-1}$ K$^{-1}$, which is significantly faster than those of other modes (about 0.01 cm$^{-1}$ K$^{-1}$), indicating it is a soft mode[34]. Theoretical studies reveal that Raman frequencies below 40 cm$^{-1}$ primarily correspond to the

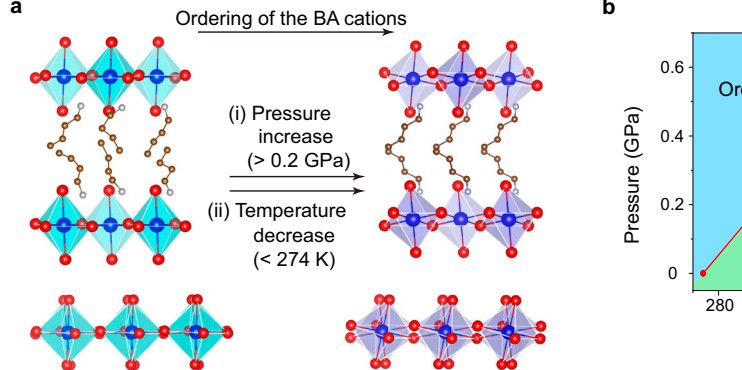
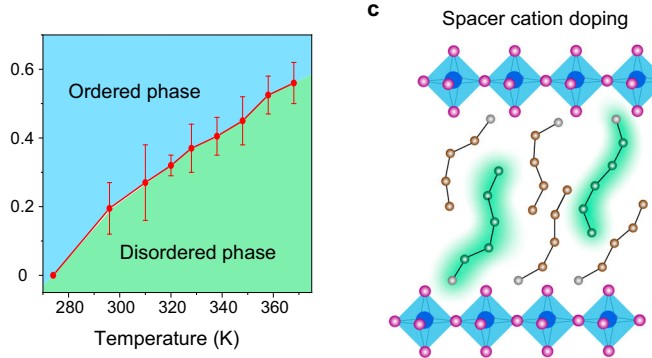

**Fig. 1 | Phase transition of (BA)$_2$PbI$_4$. a** Schematic crystal structures of ordered and disordered phase of (BA)$_2$PbI$_4$. **b** Phase diagram of (BA)$_2$PbI$_4$ as functions of temperature and pressure. The range represented by the red line is the mean value ± one standard deviation. **c** Schematic illustration of spacer cation doping in the crystal structure of (BA)$_2$PbI$_4$.

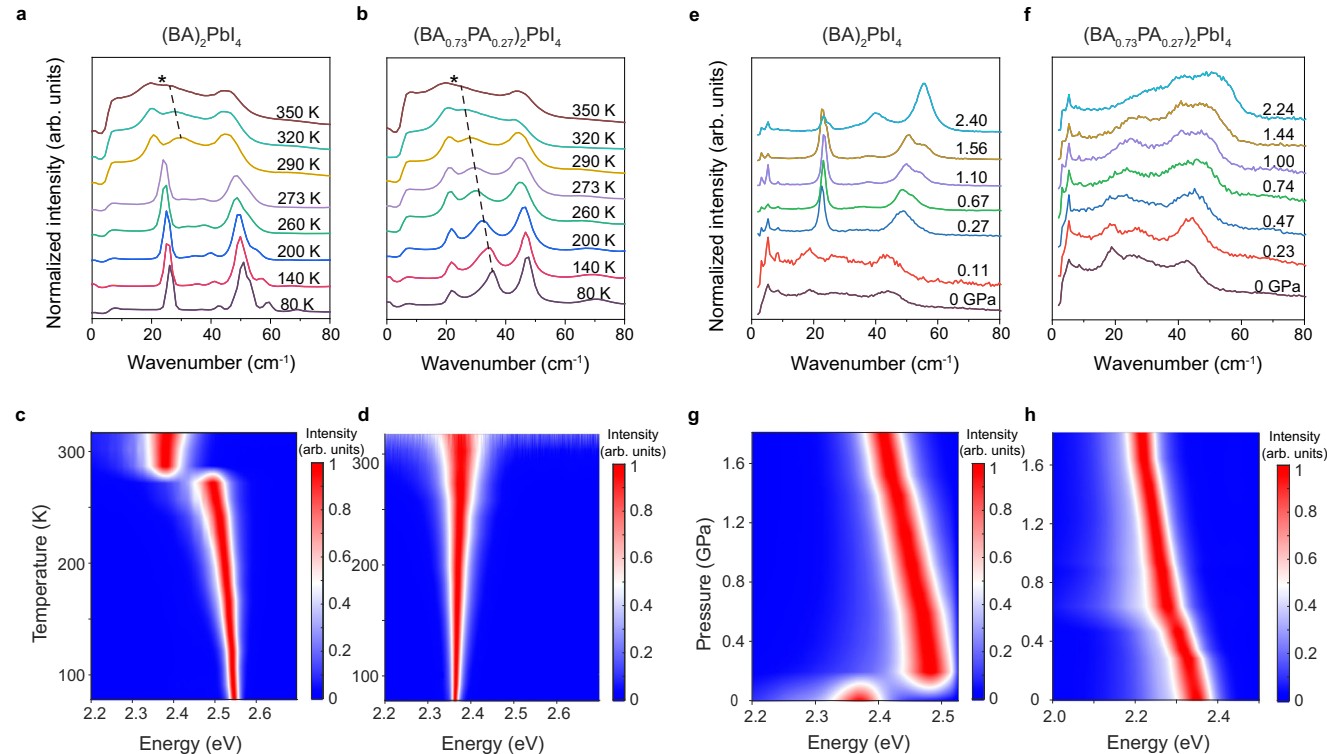

**Fig. 2 | Dopant-induced pining of the disordered phase under pressure and temperature tuning in (BA)$_2$PbI$_4$.** Temperature-dependent low-frequency Raman spectra of **a** (BA)$_2$PbI$_4$ and **b** (BA$_{0.73}$PA$_{0.27}$)$_2$PbI$_4$. Temperature-dependent PL spectra of **c** (BA)$_2$PbI$_4$ and **d** (BA$_{0.73}$PA$_{0.27}$)$_2$PbI$_4$. Pressure-dependent low-frequency

Raman spectra of **e** (BA)$_2$PbI$_4$ and **f** (BA$_{0.73}$PA$_{0.27}$)$_2$PbI$_4$. Pressure-dependent PL spectra of **g** (BA)$_2$PbI$_4$ and **h** (BA$_{0.73}$PA$_{0.27}$)$_2$PbI$_4$. The temperature-dependent evolution of vibration mode of (BA$_{0.73}$PA$_{0.27}$)$_2$PbI$_4$ at 35 cm$^{-1}$ at 80 K is marked as dashed line and the peak shoulder is marked as *.

movement of the terminal iodide atoms, characterized by the twisting around the octahedral axis. The soft mode, discerned through analysis of its polarization dependence, is attributed to the synergy of tilt and twist of PbI$_6$ octahedron[32]. Since this mode is present in the disordered phase but absent in the ordered phase, we suggest it is related to the out-of-plane tilting of the octahedra, aligning with the theoretical calculations[28,32].

The single-crystal structures of (BA)$_2$PbI$_4$ and (BA$_{0.73}$PA$_{0.27}$)$_2$PbI$_4$ show that the thermal ellipsoids of the iodide ions have severely oblate shapes. (BA$_{0.73}$PA$_{0.27}$)$_2$PbI$_4$ exhibits larger atomic displacement parameters of the lead and iodide atoms as compared to (BA)$_2$PbI$_4$ at 80 K (Supplementary Fig. 3), which can be attributed more static disorder. The persistence of the disordered phase at low temperatures can be understood in terms of a larger conformational entropy in the (BA$_{0.73}$PA$_{0.27}$)$_2$PbI$_4$, which inhibits the phase transition from a liquid-like disordered state to solid-like ordered state[35]. This behavior is similar to the phenomena in alkane mixtures, in which mixing lowers the melting temperature of the solid-liquid transition compared to the pure alkanes[36,37].

Temperature-dependent PL studies reveal distinctive optical properties between the disordered and ordered phases. The PL spectra of (BA)$_2$PbI$_4$ show a sudden change in the peak energy by 0.12 eV across the phase transition (Fig. 2c). Band structure calculations show the valence band maximum of (BA)$_2$PbI$_4$ primarily consists of a mixture of Pb 6$s$ and equatorial I 6$p$ orbitals, and the conduction band minimum derives primarily from lead 6$p$ orbitals[38–40]. Compared to the disordered phase, the larger bandgap in the ordered phase (BA)$_2$PbI$_4$ can be attributed to a larger octahedral tilting, which decreases the orbital overlap between Pb 6$s$ and equatorial I 6$p$, leading to a decreased bandwidth. By comparison, the PL spectra of (BA$_{0.73}$PA$_{0.27}$)$_2$PbI$_4$ (Fig. 2d) show the absence of a drastic change in the PL peak energy, confirming the suppression of the phase transition.

Moreover, the doping structure exhibits a slight blueshift of the PL peak energy as the temperature increases (i.e., average $(\partial E_g/\partial T)_p$ = 0.09 meV K$^{-1}$), which is similar to the disordered phase of (BA)$_2$PbI$_4$. However, this bandgap variation trend is opposite to that observed in the ordered phase of (BA)$_2$PbI$_4$, which exhibits a monotonic redshift of the PL peak energy as the temperature increases (i.e., average $(\partial E_g/\partial T)_p$ = −0.26 meV K$^{-1}$). The opposite temperature dependence of the bandgap indicates the energy renormalization of the electronic band structure caused by exciton-phonon coupling are different in the two phases[41]. Theoretical calculations have shown that the exciton-phonon coupling is susceptible to the dynamic out-of-plane octahedral tilting which modulates the quantum-well thickness and electronic bandwidths[42]. With increasing temperature, the ordered BA cations in the low-temperature phase of (BA)$_2$PbI$_4$ are becoming disordering, resulting in the gradual unlocking of the octahedra. This leads to a smaller out-of-plane octahedral tilting angle and thus a decrease of bandgap with increasing temperature.

We further investigated the exciton binding energies of (BA)$_2$PbI$_4$ in both ordered and disordered phases and a series of (BA$_{1-x}$PA$_x$)$_2$PbI$_4$ through their absorption spectra (Supplementary Fig. 4). The exciton energy was directly obtained from the absorption peak position, while the band gap energy was derived using the Tauc plot method. At room temperature, the doped structures (BA$_{0.73}$PA$_{0.27}$)$_2$PbI$_4$ and disordered phase (BA)$_2$PbI$_4$ exhibit similar exciton binding energies, ~0.38 eV, aligning with their isostructural phases. By comparison, the ordered phase (BA)$_2$PbI$_4$ at 80 K exhibits an exciton binding energy of around 0.34 eV.

We next discuss the dopant-induced pinning of the disordered phase under high pressures and its influence on optical properties. We measured the pressure-dependent Raman (Fig. 2e, f) and PL spectra (Fig. 2g, h) of (BA)$_2$PbI$_4$ and (BA$_{0.73}$PA$_{0.27}$)$_2$PbI$_4$ at room temperature. With increasing pressure, the Raman spectra of (BA)$_2$PbI$_4$ exhibit an

abrupt change from broad diffusive peaks to well-resolved narrow peaks at 0.2 GPa, which is reminiscent of that observed in the temperature-dependent studies (Fig. 2a). The Raman peaks of the high-pressure phase are consistent with those of the low-temperature phase, and they gradually shift to higher in energy with increasing pressure. By comparison, the doping structure $(BA_{0.73}PA_{0.27})_2PbI_4$ exhibits no abrupt change in the Raman spectra, indicating the absence of the pressure-induced phase transition.

Pressure-dependent PL spectra of $(BA)_2PbI_4$ shows a sudden blueshift of the PL peak energy by 0.11 eV across the phase transition at 0.2 GPa. With further pressurization, the PL peak of the ordered phase consistently redshifts from 2.48 eV to 2.39 eV, attributable to the compression of Pb-I bond. By comparison, the PL peak of $(BA_{0.73}PA_{0.27})_2PbI_4$ continuously redshifts from 2.36 eV to 2.22 eV with increasing pressure from 0 GPa to 1.82 GPa, showing the absence of phase transition. The average $(\partial E_g/\partial p)_T$ under compression is about 76 meV GPa$^{-1}$, which is larger than that of the high-pressure ordered phase of $(BA)_2PbI_4$ (i.e., 50 meV GPa$^{-1}$). In general, $(BA_{0.73}PA_{0.27})_2PbI_4$ exhibits broader PL peaks, consistent with the more disordered structure.

## Lateral homojunctions with tunable optical properties

As the doping structure is pinned in the disordered phase under high pressures or low temperatures, this provides a unique opportunity to construct lateral homojunction with distinct electronic structures by pressure or temperature tuning. Since the interlayer distance is only slightly increased with a small quantity of dopant (Supplementary

Fig. 5), we achieved lateral epitaxial growth of $(BA)_2PbI_4 - (BA_{1-x}PA_x)_2PbI_4$ homojunctions by introducing PA dopant in the crystal growth of $(BA)_2PbI_4$ (Fig. 3a). Firstly, we grown single-crystal microplates of $(BA)_2PbI_4$ on the air-solution interface of a precursor solution droplet[15]. Subsequently, another saturated solution droplet containing mixed $(BA)_2PbI_4$ and $(PA)_2PbI_4$ was introduced into the droplet, leading to epitaxial growth of $(BA_{1-x}PA_x)_2PbI_4$ on the edges of the existing $(BA)_2PbI_4$ microplates. The second step maintained a supersaturation condition such that the redissolution of the pre-grown $(BA)_2PbI_4$ was suppressed. The compositions of the later-grown $(BA_{1-x}PA_x)_2PbI_4$ can be estimated from the interlayer spacing distance determined by powder X-ray diffraction (PXRD) (Supplementary Fig. 5). We monitored the epitaxial growth process of a $(BA)_2PbI_4 - (BA_{0.8}PA_{0.2})_2PbI_4$ homojunction under the optical microscope (Supplementary Fig. 6). The junctions can be visualized by the two regions with different optical contrast and a clear interface (Fig. 3b). Atomic force microscope (AFM) image reveals a step-terrace morphology on the surface (Fig. 3c). More importantly, it is clearly shown that the 2D perovskite layers maintain the integrity and the same orientation across the homojunction, indicating coherent epitaxial growth.

We demonstrate effective tuning of the optical and structural properties of the homojunction under pressure and temperature stimuli (Fig. 3d). Figure 3e–g shows the PL peak energy mapping on a homojunction under different pressures at room temperature and the difference in the PL peak energy across the interface is provided in Fig. 3h. The two regions across the interface exhibit almost identical PL spectra under ambient conditions (Fig. 3i), since both $(BA)_2PbI_4$ and

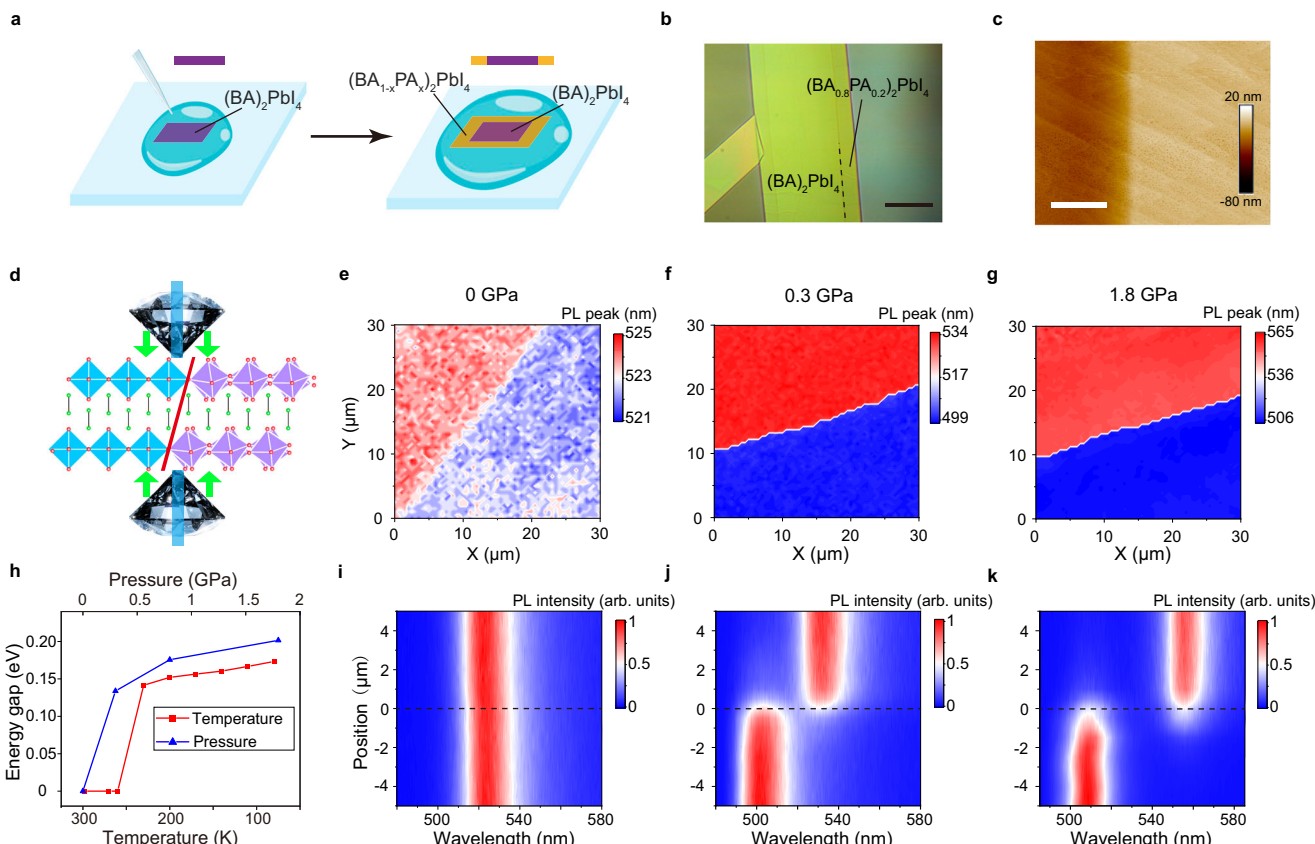

**Fig. 3 | Epitaxial growth of $(BA)_2PbI_4-(BA_{1-x}PA_x)_2PbI_4$ lateral homojunctions and their tunable optical properties under pressure and temperature stimuli.** **a** Schematic illustration of the growth process of $(BA)_2PbI_4-(BA_{1-x}PA_x)_2PbI_4$ homojunctions. **b** Optical image of a $(BA)_2PbI_4-(BA_{0.8}PA_{0.2})_2PbI_4$ homojunction. Scale bar = 50 μm. **c** Atomic force microscope image of a $(BA)_2PbI_4-(BA_{0.8}PA_{0.2})_2PbI_4$. Scale bar = 2 μm. **d** Illustration of the formation of the

junction with distinctive octahedral tilting by applying pressure. Mapping of the PL peak position of a $(BA)_2PbI_4-(BA_{0.8}PA_{0.2})_2PbI_4$ with an applied pressure of **e** 0 GPa, **f** 0.3 GPa, and **g** 1.8 GPa. **h** The difference in the PL peak energy across the interface as functions of temperature and pressure. The corresponding spatially resolved PL spectra of the homojunction along the direction across the interface with an applied pressure of **i** 0 GPa, **j** 0.3 GPa, and **k** 1.8 GPa.

$(BA_{0.8}PA_{0.2})_2PbI_4$ adopt the same disordered phase. With increasing pressure to 0.3 GPa, distinct PL spectra across the interface appear due to the occurrence of the phase transition to the ordered phase in the $(BA)_2PbI_4$ region (Fig. 3j). The difference in the PL peak energy between the two regions is 0.10 eV, which is close to the amount of bandgap shift induced by the phase transition in $(BA)_2PbI_4$. Previous studies have shown that the octahedral tilting mainly causes a downshift in the valence band maximum, while there is no obvious shift in the conduction band minimum[43]. Therefore, a close to type–I band alignment is expected under pressure. Because the two phases exhibit different pressure dependence of the bandgap, further compression effectively modulates the energy diagram of the interface. For example, when the pressure is increased to 1.8 GPa, the PL peak energy difference increases to 0.20 eV (Fig. 3k). Similar effects on modulating the optical properties can be realized by temperature tuning (Fig. 3h). Supplementary Fig. 7 shows the spatially resolved PL spectra across the interface at various temperatures. The difference in the PL peak energy across the interface can be tuned from 0 eV to 0.17 eV.

The dopant-induced phase pinning has also been observed in $(PA)_2PbI_4$ (Fig. 4a) and can be further applied to fabricate homojunctions with gradient optical properties under ambient conditions (Fig. 4b). The structure of $(PA)_2PbI_4$ exhibits an order-disorder phase transition with a transition temperature of ~315 K[10]. Therefore, an important difference between $(PA)_2PbI_4$ and $(BA)_2PbI_4$ at room temperature is that $(PA)_2PbI_4$ crystallizes in the ordered phase while $(BA)_2PbI_4$ crystallizes in the disordered phase. Compared to $(BA)_2PbI_4$, the phase transition of $(PA)_2PbI_4$ further exhibits a relative shift of the inorganic layers.

Temperature-dependent Raman spectra of $(PA)_2PbI_4$ reveal qualitatively similar lattice dynamics to $(BA)_2PbI_4$ (Supplementary Fig. 8). However, at low temperatures, an additional peak splitting in $(PA)_2PbI_4$

is observable in comparison to $(BA)_2PbI_4$, particularly at the mode at around 26 cm⁻¹ (Supplementary Fig. 9), due to a lower crystal symmetry. Temperature-dependent PL studies revealed that $(PA)_2PbI_4$ exhibits a continuous redshift of the PL peak energy across the phase transition (Fig. 4c), in contrast to a sudden change in the PL peak energy of $(BA)_2PbI_4$ and a nearly stable PL peak of doped $(BA_{0.18}PA_{0.82})_2PbI_4$ (Fig. 4d). Moreover, when the structure approaches the phase transition, the change of the PL peak energy becomes more prominent. To understand the origin, we analyzed the evolution of the single-crystal structure with temperature (Supplementary Table 1). The I–Pb–Pb–I dihedral angle for the equatorial iodides becomes smaller with increasing the temperature, indicating a decreased out-of-plane octahedral tilting. This dihedral angle drastically changes when the low-temperature ordered phase approaches the phase transition, which correlates well with the temperature dependence of the PL peak energy (Fig. 4e). The atomic displacement parameters of the organic cations and iodides also drastically increase near the transition temperature (Supplementary Fig. 10), suggesting the phase transition is driven by the disordering of the organic cations and the unlocking of the octahedral tilting. These results are consistent with the significant broadening of the Raman peaks across the phase transition.

We found the high-temperature disordered phase of $(PA)_2PbI_4$ can be stabilized at room temperature by doping with a small amount of BA cation (>10%). Single-crystal XRD studies show $(BA_{0.18}PA_{0.82})_2PbI_4$ adopts the same orthorhombic phase *Pbca* at room temperature and 80 K. The resistance to phase transition can also be seen in the temperature-dependent Raman (Supplementary Fig. 8) and PL spectra (Fig. 4d) of $(BA_{0.18}PA_{0.82})_2PbI_4$, which show absence of an abrupt change with temperature. We then synthesized lateral homojunction of $(PA)_2PbI_4$–$(BA_{1-x}PA_x)_2PbI_4$ using the above-described two-step sequential growth method. The dopant ratio in $(BA_{1-x}PA_x)_2PbI_4$

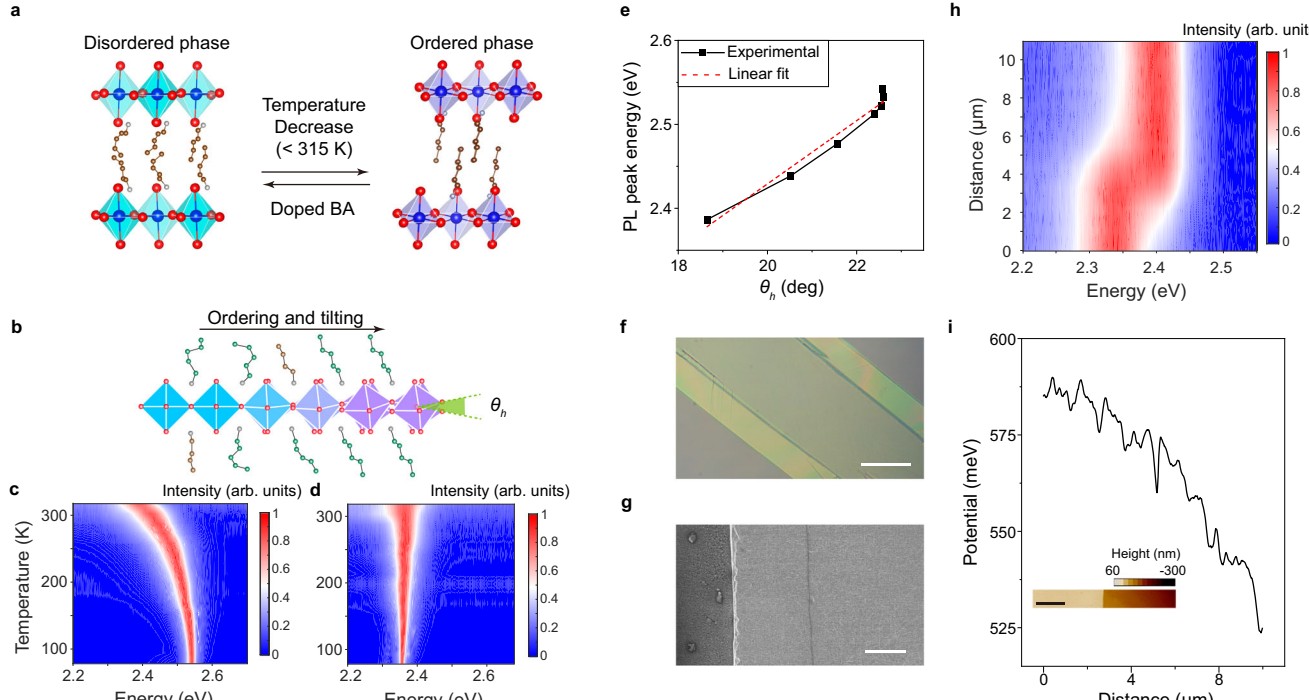

**Fig. 4 | $(PA)_2PbI_4$–$(BA_{1-x}PA_x)_2PbI_4$ homojunctions with gradient properties under ambient conditions. a** Schematic crystal structures and phase transition of $(PA)_2PbI_4$. **b** Schematic illustration of $(PA)_2PbI_4$–$(BA_{1-x}PA_x)_2PbI_4$ homojunction with gradient out-of-plane octahedral tilting. The extent of octahedral tilting is characterized by horizontal I-Pb-Pb-I dihedral angle $\theta_h$. Temperature-dependent PL spectra of **c** $(PA)_2PbI_4$ and **d** $(BA_{0.18}PA_{0.82})_2PbI_4$. **e** Correlation between the PL peak energy and the horizontal dihedral angle ($\theta_h$) in $(PA)_2PbI_4$ at various temperatures. **f** Optical image of $(PA)_2PbI_4$–$(BA_{0.2}PA_{0.8})_2PbI_4$. Scale bar = 50 μm. **g** SEM image of $(PA)_2PbI_4$–$(BA_{0.2}PA_{0.8})_2PbI_4$. Scale bar = 10 μm. **h** Spatially resolved PL spectra of a $(PA)_2PbI_4$–$(BA_{0.2}PA_{0.8})_2PbI_4$ homojunction across the interface under ambient condition. **i** The relative working function across the interface measured by Kelvin probe force microscopy in a $(PA)_2PbI_4$–$(BA_{0.2}PA_{0.8})_2PbI_4$ homojunction. The inset is the atomic force microscope topography, showing a height step at the interface. Scale bar = 2 μm.

were determined by PXRD using Vegard's Law. When a small BA dopant ratio was used in the second step growth, a relative smooth interface was formed because of a small lattice mismatch (Fig. 4f, g). With increasing the BA dopant ratio, periodic cracks were observed in the outer region (Supplementary Fig. 11). We take this as an indicator of strain relaxation because of a large in-plane lattice mismatch in this junction.

High-resolution PL spectroscopic mapping on a $(PA)_2PbI_4$–$(BA_{0.2}PA_{0.8})_2PbI_4$ homojunction reveal a gradient transition from the ordered phase to disordered phase across the interface under ambient conditions (Fig. 4h). The spatially resolved PL spectra exhibit a gradual redshift from the $(PA)_2PbI_4$ region (2.37 eV) to $(BA_{0.2}PA_{0.8})_2PbI_4$ region (2.42 eV). The bandgap variation is similar to that near the phase transition of $(PA)_2PbI_4$ with increasing temperature (Fig. 4c), indicating the degree of octahedral tilting decreases spatially across the interface. The spatial dependence of the octahedral tilting in the homojunction can be inferred from the PL peak position based on the established correlation in Fig. 4e. The difference in the PL peak energy is consistent with the gradual change of working function (~70 meV) across the interface measured by Kelvin probe force microscopy (Fig. 4i). Moreover, the Raman spectra taken at multiple locations across the interface also exhibit a continuous change in the vibrational modes associated with both the inorganic framework and the organic cation (Supplementary Fig. 12).

### Directional exciton diffusion in the homojunction

We investigated exciton diffusion in a $(BA)_2PbI_4$–$(BA_{0.8}PA_{0.2})_2PbI_4$ homojunction at 80 K using PL imaging spectroscopy. A continuous wave laser beam was tightly focused to the diffraction limit to locally generate excitons and the PL intensity near the excitation spot region was collected to show the diffusion of exciton. Figure 5a–d shows the PL images when the excitation laser was focused on $(BA)_2PbI_4$ (Fig. 5a), $(BA_{0.8}PA_{0.2})_2PbI_4$ (Fig. 5b), and the interface (Fig. 5c), and the reflected laser beam image provided in Fig. 5d. The broader PL spot relative to the laser spot arises from exciton diffusion. The spatial profile of PL intensity is broader in the disordered phase than that in the ordered phase (Fig. 5e), indicating a faster exciton diffusion in the former. This is because the disordered phase exhibits more dispersive bands due to a smaller octahedral tilting, thus a lower effective exciton mass. We extracted the exciton diffusion lengths by fitting the modified Bessel

function $I(r) = \alpha + \beta k_0(r/L)$, where $r$ is the radical distance, $L$ is the diffusion length, $\alpha$, $\beta$, and $k_0$ are the fitting parameters[44]. The obtained exciton diffusion lengths were $1.2 \pm 0.1$ and $0.8 \pm 0.1$ μm for the disordered and ordered phase (Fig. 5e), respectively. Figure 5f shows the spatial profiles of the PL intensity for diffusion perpendicular to and parallel to the interface with the excitation laser being focused on the interface. On the disordered phase, the exciton diffusion length perpendicular to the interface was $1.6 \pm 0.1$ μm, which is slightly longer than that parallel to the interface, indicating directional carrier funneling from the ordered phase to disordered phase. Asymmetric exciton diffusion across the interface can also be visualized when the excitation laser was focused on either the ordered or disordered phase near the interface (Fig. 5a, b).

In summary, our work has revealed that spacer cation doping prevents the ordering transition of the spacer cation in 2D LHPs at low temperatures or under high pressure conditions, resulting in the stabilization of the disordered phase in a wider range of thermodynamic conditions. Leveraging this discovery, we have successfully fabricated lateral homojunctions composed of ordered and disordered phases of 2D LHPs through epitaxial growth by introducing dopant in the crystal growth process. We have further demonstrated effective control over the optical properties of the homojunction across the interfaces, as the two phases exhibit distinctive temperature and pressure dependences of the electronic band structures. Moreover, directional exciton diffusion across the interface was observed. Our results demonstrate spacer cation engineering as an effective approach for constructing 2D homojunctions, providing exciting opportunities for investigating emergent properties at the interface through thermodynamic tuning.

## Methods
### Starting materials
PbI₂ (98%, Macklin), $n$-butylamine (99.5%, Sigma-Aldrich), $n$-amylamine (99.0%, Sigma-Aldrich), HI (47% wt% in $H_2O$, stabilized with 1.5% $H_3PO_2$, Macklin) and $H_3PO_2$ (50 wt% in $H_2O$, Macklin) were obtained commercially and used without further purification.

### Synthesis of $(BA_{1-x}PA_x)_2PbI_4$
To synthesize single-crystal $(BA_{1-x}PA_x)_2PbI_4$, PbI₂ powder (0.461 g, 1.0 mmol), $n$-butylamine, and $n$-amylamine were added in a solution

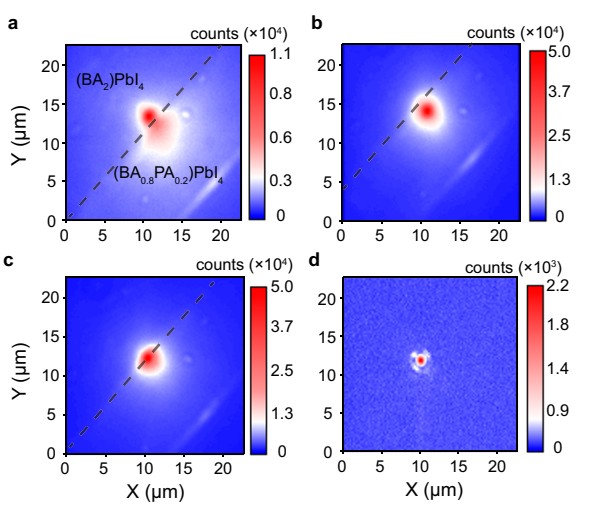
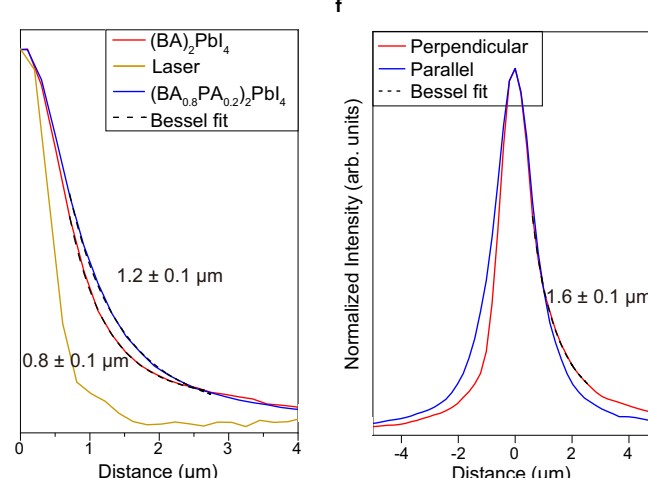

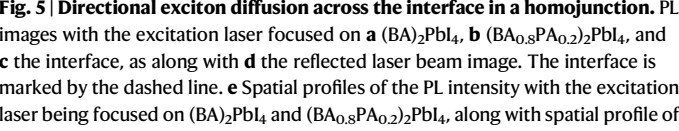

**Fig. 5 | Directional exciton diffusion across the interface in a homojunction.** PL images with the excitation laser focused on **a** $(BA)_2PbI_4$, **b** $(BA_{0.8}PA_{0.2})_2PbI_4$, and **c** the interface, as along with **d** the reflected laser beam image. The interface is marked by the dashed line. **e** Spatial profiles of the PL intensity with the excitation laser being focused on $(BA)_2PbI_4$ and $(BA_{0.8}PA_{0.2})_2PbI_4$, along with spatial profile of the reflected laser intensity. The data are extracted along the direction parallel to the interface and through the center of the laser spot. The dash line is the corresponding fit. **f** Spatial profiles of the PL intensity extracted from **c** for diffusion perpendicular to and parallel to the interface direction. The dash line is the corresponding fit.

comprising 7.0 ml of concentrated hydroiodic acid and 0.5 ml of hypophosphorous acid. The solution was heated and stirred on a hot plate at 100 °C until all the reagents were dissolved. Subsequently, the solution was allowed to cool to room temperature naturally, resulting in precipitation of orange plate-shaped crystals. Specifically, for the synthesis of $(BA_{0.73}PA_{0.27})_2PbI_4$, 1.5 mmol of *n*-butylamine and 0.5 mmol of *n*-amylamine were added. For the synthesis of $(BA_{0.40}PA_{0.60})_2PbI_4$, 1.0 mmol *n*-butylamine and 1.0 mmol *n*-amylamine were added. For the synthesis of $(BA_{0.18}PA_{0.82})_2PbI_4$, 0.5 mmol of *n*-butylamine and 1.5 mmol of *n*-amylamine were added. For the synthesis of $(BA)_2PbI_4$ and $(PA)_2PbI_4$, pure *n*-butylamine (2.0 mmol) and *n*-amylamine were added, respectively. The doping ratio in $(BA_{1-x}PA_x)_2PbI_4$ was determined by $^1H$ NMR, with spectra obtained using a Bruker 400 MHz system (Supplementary Figs. 13–17). The crystals were dissolved in DMSO-d$_6$.

## Synthesis of homojunction

$(BA)_2PbI_4$–$(BA_{1-x}PA_x)_2PbI_4$ homojunctions were synthesized by a two-step floating growth method. For the synthesis of $(BA)_2PbI_4$–$(BA_{0.8}PA_{0.2})_2PbI_4$, a droplet of saturated $(BA)_2PbI_4$ solution and a droplet of saturated $(BA_{0.73}PA_{0.27})_2PbI_4$ solution were first introduced side by side onto a glass slide. The solution was spread out to reduce the curvature of the drop and microplate crystals crystallized on the air-solution interface. Subsequently, the two droplets were merged using a needle. The merged solution was left for -10–20 min for allowing further growth of the doping phase on the existing $(BA)_2PbI_4$ crystals. $(PA)_2PbI_4$–$(BA_{1-x}PA_x)_2PbI_4$ homojunctions were synthesized using a similar method. In the synthesis of $(PA)_2PbI_4$–$(BA_{0.2}PA_{0.8})_2PbI_4$, $(PA)_2PbI_4$–$(BA_{0.5}PA_{0.5})_2PbI_4$, and $(PA)_2PbI_4$–$(BA_{0.6}PA_{0.4})_2PbI_4$, a droplet of saturated $(PA)_2PbI_4$ solution was introduced onto the glass slide together with a drop of a saturated solution of $(BA_{0.18}PA_{0.82})_2PbI_4$, $(BA_{0.40}PA_{0.60})_2PbI_4$, and $(BA_{0.73}PA_{0.27})_2PbI$, respectively. The doping ratios in the homojunctions can be estimated using PXRD collected on the crystals that precipitated out in the merged solution. Homojunctions were picked-up by contacting the crystals with a polydimethylsiloxane film (PDMS, from GelPak, PF-30-X4) and subsequently transferred onto various substrates, such as Si/SiO$_2$ and copper tape, through pressing the PDMS film on the substrates[15].

## Structural characterizations

The powder X-ray diffraction data were collected using an X-Pert3 powder X-ray diffractometer with Cu Kα radiation ($\lambda = 1.54056$ Å). Temperature-dependent single-crystal X-ray diffraction measurements on $(PA)_2PbI_4$ and various $(BA_{1-x}PA_x)_2PbI_4$ were performed using an XtaLAB PRO 007HF (Mo) single-crystal X-ray diffractometer with Mo Kα radiation ($\lambda = 0.71073$ Å). The structures were solved by charge flipping and refined by full-matrix least-squares on F2 using the Jana 2006 package. The CCDC reference number of $(BA_{1-x}PA_x)_2PbI_4$ at 293 K and 80 K is 2283140, 2283141, 2283145, 2283146, 2283147 and 2283149, respectively.

The as-grown homojunctions were picked up by a PDMS film and then transferred to the Si/SiO$_2$ substrate for morphology characterization using optical microscope (AOSVI), field-emission scanning electron microscope (S-4800, Hitachi), and atomic force microscope (AFM) (Dimension Icon, Bruker). SEM images were collected with an accelerating voltage of 2 kV. The AFM and KPFM measurements were carried out by a Bruker Dimension Icon apparatus using the ScanAsyst-Air mode with SCANASYST-AIR probes (tip radius: 2 nm) and the KPFM-AM mode with PFQNE-AL probes (tip radius: 5 nm), respectively. The PeakForce setpoint was set to below 500 pN to minimize the interaction between the sample and probes. In the KPFM-AM procedure, the lift height was decreased to 50 nm during the interleave scan to enhance resolution. The absolute work function of probes was not calibrated.

## Optical characterizations

Temperature-dependent PL spectra measurements on the various samples were conducted on a homebuilt microscope in a reflection geometry. The measurements were performed from 80 K to 320 K by using a cryostat (Linkam THMS600). A stabilized 405 nm continuous-wave laser (TEM-F-405 nm) was used to excite the samples with a power of about 100 nW at room temperature and 500 nW at low temperature. At 80 K, the samples retained stable under 405 nm laser exposure with a power of 5 μW for 30 min (Supplementary Fig. 18), indicating that laser-induced damage has minimal impact on the experiments. The laser beam was focused onto the samples by a ×50 objective (NA 0.4, Nikon) with long working distance. The PL emission was detected by a spectrograph (Princeton Instruments, HRS-300S) and a liquid-nitrogen-cooled charge-coupled device camera (Princeton Instruments, PyLoN-400BRX). The exciton diffusion and spatially resolved PL measurements on the $(BA)_2PbI_4$–$(BA_{1-x}PA_x)_2PbI_4$ homojunction at various temperatures were performed using the same microscope and a cryostat (Physike Scryo-S-300MS). The 405 nm laser was focused by a ×40 objective (NA 0.60, Olympus). To image the PL emission or the reflected laser, a 425 nm long-pass dichroic filter or a $390 \pm 20$ nm bandpass filter was placed in front of the spectrometer, respectively. For the spatially and spectrally resolved PL measurement, a convex lens was placed in front of the objective to expand the laser excitation spot and the PL spectra were acquired along the entrance slit of the spectrometer. The PL mapping of $(PA)_2PbI_4$–$(BA_{1-x}PA_x)_2PbI_4$ homojunction under ambient temperature was obtained by using a Q2 laser scanning confocal microscope (ISS) with a 405 nm laser as the excitation source. The excitation laser was focused by a ×40 objective (NA 0.95, Nikon) and a 414 nm-long pass dichroic filter was used to reflect the excitation light before spectrograph. The PL emission was detected by a spectrograph (iHR320, Horiba Scientific) and a CCD camera (Syncerity, Horiba Scientific). The absorption spectra of the samples were measured in a transmission geometry with a stabilized tungsten light source (Thorlabs, SLS201L(/M)). The samples were prepared as thin sheets by mechanical exfoliation.

Temperature-dependent Raman spectra were collected using the same homebuilt microscope, with a 633 nm He–Ne laser (NLHP-925, Newport) serving as the excitation source. To eliminate the undesirable amplified spontaneous emission, the laser beam was passed through a volume Bragg grating (VBG) bandpass filter (Optigrate, linewidth: 5 cm$^{-1}$) before entering the microscope. To minimize the Rayleigh scattered light, the collected signal was passed through the same VBG filter and subsequently filtered by three VBG notch filters (Optigrate, OD > 3) before entering the spectrograph. Line-scan Raman spectra shown in Supplementary Fig. 12 were measured using a Jobin Yvon LabRam HR 800 micro-Raman spectrometer with an excitation wavelength of 633 nm.

Pressure-dependent PL measurements were performed on a home-built spectroscopy system in a reflection geometry. The pressure was controlled by diamond anvil cells, employing Type IIa ultra-low fluorescence diamonds with a culet size of 500 μm. The sample chamber was constructed using a pre-indented T301 stainless-steel gasket, -50 μm thick, with a laser-drilled hole of about 300 μm. Silicone oil served as the pressure-transmitting medium. The samples and a ruby ball (for pressure measurements) were loaded inside the chamber, and the pressures were measured by the ruby fluorescence method[45]. A stabilized 488 nm continuous-wave laser (06-01, Cobalt) was used as the excitation source, which was focused onto the sample by a long working distance ×100 objective (NA 0.55, Mitutoyo). The emission was collected by the same objective and detected by a spectrograph (Princeton Instruments, HRS-300) and a charge-coupled device camera (Princeton Instruments, PIXIS: 100BR_eXcelon). The PL mapping of homojunction under different pressures was obtained with the same microscope. The position of homojunction was

controlled using a scanning stage equipped with an integrated measuring system (SCANplus 75 × 50, Marzhauser Wetzlar). Pressure-dependent low-frequency Raman spectra were collected using a homebuilt microscope with a 633 nm laser (08-01, Cobolt). The Raman signals were detected by a spectrograph (Princeton Instruments, Iso-Plane 320) and a liquid-nitrogen-cooled charge-coupled device camera (Princeton Instruments, PyLon100BR_eXcelon).

## Data availability

Crystallographic data for the structures reported in this Article have been deposited at the Cambridge Crystallographic Data Centre, under deposition numbers CCDC 2283140, 2283141, 2283145, 2283146, 2283147 and 2283149. Copies of the data can be obtained free of charge via https://www.ccdc.cam.ac.uk/structures/. Source data are provided with this paper.

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

## Acknowledgements

This work is supported by the National Natural Science Foundation of China 22271006 (Y.F.), 22275004 (X.-J.L.), 8200906190 (X.-J.L.) and the Fundamental Research Funds for the Central Universities, Peking University. The measurements of H$^1$NMR, SCXRD, PXRD and multi-dimensional confocal microfluorescence imaging system were performed at the Analytical Instrumentation Center of Peking University.

## Author contributions

Y.F. and X.-J.L. conceived the idea. H.H. synthesized the samples and performed the optical characterizations with the help of L.J., J.G., S.C., X.H., and Y.G. S.G. performed the high-pressure studies with the assistance of Y.M. X.L. solved the single-crystal structures. Y.C. and Y.L. performed AFM and KFM measurements. Y.F. and X.-J.L. supervised the project. Y.F., H.H., S.G., and X.-J.L. wrote the manuscript. All authors have interpreted the findings, commented on the paper, and approved the final version.

## Competing interests

The authors declare no competing interests.
