## [Peer Review File · Nature Communications]

Two-Dimensional Lead Halide Perovskite Lateral Homojunctions Enabled by Phase PinningREVIEWER COMMENTS

Reviewer #1 (Remarks to the Author):

2D lead halide perovskites are important soft semiconductors attractive for many applications. Their properties are affected by order-disorder phase transitions and this feature can be used for construction of homojunctions comprising the same material with distinct structural phases. The main problem for construction of such homojunctions is stability of different polymorphs at ambient conditions. In the submitted manuscript, the authors showed that doping n-butylammonium (BA⁺) lead iodide with n-pentylammonium (PA⁺) cations (or vice versa) stabilizes the high-temperature disordered phases. Interestingly, the doped samples show presence of the disordered phase down to at least 80K. I like this idea very much and in my opinion this study shows an efficient way for obtaining new phases with suppressed phase transitions, which are attractive for applications. There are many studies on 3D lead halide perovskites reporting benefits of mixed-cation approach and presence in some cases new phases, like dipolar glass, but studies of mixed-organic cation in 2D lead halide perovskites are very scarce. I expect, therefore, that this paper will attract broad interest of the scientific community and I support publication of these results in Nature Communications. However, presentation needs major revision according to the following points:

1. Introduction: information is given that temperature and pressure may cause dramatic changes in the optical and electronic properties of 2D perovskites. One more reference should be added which reports unusual mechanism of the pressure-induced mechanism in a 2D lead bromide (DOI: 10.1021/acs.chemmater.2c01533).
2. The authors are not consistent in the names of the compounds and reagents since they use n-propylamine or amylamine and (BA_{1-x}PA_x)₂PbI₄ or (PA_{1-x}BA_x)₂PbI₄ – the names should be consistent in the whole text and supplementary material.
3. The authors provided cifs of mixed-cation systems but these cifs assume presence of only PA⁺ cations. I understand that presence of both organic cations cannot be modeled but why cif of (BA_{0.73}PA_{0.27})₂PbI₄, where amount of BA⁺ is about 3 times larger than PA⁺, is also modeled assuming only PA⁺ cations?
4. DSC data of the studied compounds should be added to prove suppression of the phase transitions.
5. Raman: low-frequency Raman spectra in Fig. 2a should also be shown in a traditional way, as in the HP spectra in Fig. 2c. This way of presentation would better show strong broadening and central peak. The authors also wrote “the Raman modes for both inorganic cages (Fig. 2a) and organic cations (Fig. S1) in the (BA_{0.73}PA_{0.27})₂PbI₄ continuously broaden with increasing temperature, showing the absence of an abrupt change.” However, figure S1 shows spectra for BA_{0.40}, not BA_{0.73}. Furthermore, the temperature is given in Celsius degree instead of Kelvin, as in other figures and tables. Please correct this figure.
6. Raman spectra: the authors noticed appearance of a new mode at 35 cm⁻¹ for (BA_{0.73}PA_{0.27})₂PbI₄, not observed for the ordered BA₂PbI₄. It seems also not present in the disordered BA₂PbI₄. I do not understand origin of this new mode since if (BA_{0.73}PA_{0.27})₂PbI₄ and disordered BA₂PbI₄ have the same orthorhombic structure, their low-frequency spectra should be very similar. Can you propose origin of this mode, which is also clearly observed for (BA_{0.18}PA_{0.82})₂PbI₄ shown in figure S6? By the way, please show Raman spectra of the PA and PA_{0.82} samples in Fig. S6 in the same frequency range to facilitate comparison. I also do not see clearly additional splitting in PA₂PbI₄ compared to BA₂PbI₄ – please add Raman spectra presented in the traditional way to show the mentioned splitting.
7. The authors wrote “More disorder in the (BA_{0.73}PA_{0.27})₂PbI₄ can be seen by the larger

atomic displacement parameters of the organic chain and iodide (Fig. S2).” I am not convinced that this conclusion is correct because the cif does not take into account presence of two different cations but only PA⁺ (see my comment 3). Furthermore, Raman spectra of (BA_{0.73}PA_{0.27})₂PbI₄ presented in Fig. S2 show strong narrowing of bands on cooling and it seems that FWHM of the bands is slightly larger than FWHM of the corresponding bands of BA₂PbI₄ and PA₂PbI₄. This suggests static rather than dynamic disorder at 80 K for the mixed BA-PA compounds. Numerous literature reports on lead halide perovskites also showed that organic cations are well-ordered at 80 K and, therefore, static disorder in the mixed-cation samples is a more reasonable explanation.

8. Please check the text and figures for typos and errors. For instance, “decrease” in Fig. 4a. References should be corrected (subscripts, names added up to ten, abbreviation of journals etc.).

9. In the Structural Characterization and Acknowledgements information is given that NMR spectra were measured. However, no NMR spectra were presented and no information what they showed.

Reviewer #3 (Remarks to the Author):

In this work, the authors have proven a novel an interesting method to create homojunction of different 2D perovskites. By cation doping they were able to tune the ordering transition of the spacer cation and the associated out-of-plane octahedral tilting in the inorganic framework, resulting in phase pinning of the disordered phase when decreasing temperature or increasing pressure. This is an innovative concept in material engineering for tuning perovskite crystalline order and thus modifying their optoelectronic properties at atomic scale. In addition the authors have verified the exciton diffusion, which revealed interesting fundamental advance. I therefore propose to accept the work, after checking the following minor points:

1. what is the stability of the material under laser exposure for the measurements?
2. What is the excitation density used?
3. can the author retrieve the exciton binding energy?

Response to the Reviewers' Comments

We are very grateful to the Reviewers for their time and efforts to review our manuscripts. Reviewers' suggestions and criticisms are very helpful for us and we have taken all these comments into account in the revised manuscript. In the following, we reproduce the reviewers' comments in black fonts and give our responses in blue.

Reviewer: 1

Comments:

2D lead halide perovskites are important soft semiconductors attractive for many applications. Their properties are affected by order-disorder phase transitions and this feature can be used for construction of homojunctions comprising the same material with distinct structural phases. The main problem for construction of such homojunctions is stability of different polymorphs at ambient conditions. In the submitted manuscript, the authors showed that doping n-butylammonium (BA⁺) lead iodide with n-pentylammonium (PA⁺) cations (or vice versa) stabilizes the high-temperature disordered phases. Interestingly, the doped samples show presence of the disordered phase down to at least 80K. I like this idea very much and in my opinion this study shows an efficient way for obtaining new phases with suppressed phase transitions, which are attractive for applications. There are many studies on 3D lead halide perovskites reporting benefits of mixed-cation approach and presence in some cases new phases, like dipolar glass, but studies of mixed-organic cation in 2D lead halide perovskites are very scarce. I expect, therefore, that this paper will attract broad interest of the scientific community and I support publication of these results in Nature Communications. However, presentation needs major revision according to the following points:

We thank the Reviewer for the support of our work and valuable suggestions.

Question 1. Introduction: information is given that temperature and pressure may cause dramatic changes in the optical and electronic properties of 2D perovskites. One more reference should be added which reports unusual mechanism of the pressure-induced mechanism in a 2D lead bromide (DOI: 10.1021/acs.chemmater.2c01533).

We thank the Reviewer for bringing this significant work to our attention, which has been cited as reference 32.

Question 2. The authors are not consistent in the names of the compounds and reagents since they use n-propylamine or amylamine and $(\text{BA}_{1-x}\text{PA}_x)_2\text{PbI}_4$ or $(\text{PA}_{1-x}\text{BA}_x)_2\text{PbI}_4$ – the names should be consistent in the whole text and supplementary material.

We thank the Reviewer for the careful reading. We have consistently changed the format to $(\text{BA}_{1-x}\text{PA}_x)_2\text{PbI}_4$ throughout the revised manuscript.

Question 3. The authors provided cifs of mixed-cation systems but these cifs assume presence of only PA^+ cations. I understand that presence of both organic cations cannot be modeled but why cif of $(\text{BA}_{0.73}\text{PA}_{0.27})_2\text{PbI}_4$, where amount of BA^+ is about 3 times larger than PA^+ , is also modeled assuming only PA^+ cations?

We thank the Reviewer for pointing out this important aspect. For the structure of $(\text{BA}_{0.73}\text{PA}_{0.27})_2\text{PbI}_4$, we have followed the Reviewer's suggestion to use BA^+ cation in the structural modeling. The CIF of $(\text{BA}_{0.73}\text{PA}_{0.27})_2\text{PbI}_4$ has been updated. The inorganic framework remains unchanged, as the organic cations in the doped structures are indistinguishable.

Question 4. DSC data of the studied compounds should be added to prove suppression of the phase transitions.

Following the Reviewer's suggestion, we have conducted DSC measurements on $(\text{BA})_2\text{PbI}_4$, $(\text{PA})_2\text{PbI}_4$, and three doped compounds $(\text{BA}_{1-x}\text{PA}_x)_2\text{PbI}_4$. The results are shown in Fig. S1, as reproduced below. $(\text{BA})_2\text{PbI}_4$ and $(\text{PA})_2\text{PbI}_4$ exhibited endothermic peaks at 275 K and 324 K, respectively, corresponding to the order-disorder phase transition. In contrast, the three doped $(\text{BA}_{1-x}\text{PA}_x)_2\text{PbI}_4$ showed flat baselines between 235-330 K, signifying no phase transition. Additionally, we extended the DSC analysis of $(\text{BA}_{0.73}\text{PA}_{0.27})_2\text{PbI}_4$ to as low as 130 K (the lower limit of our instrument) and observed no phase transition. These results align with our findings from the temperature-dependent PL and Raman spectra. The new results and discussions have been incorporated as Figure S1 and detailed on page 5 of the revised manuscript.

Fig. S1. (a) Differential scanning calorimetry (DSC) of $(BA)_2PbI_4$, $(PA)_2PbI_4$, and three doped $(BA_{1-x}PA_x)_2PbI_4$ in the range of 235-330 K. (b) DSC of $(BA_{0.73}PA_{0.27})_2PbI_4$ in the range of 130-330 K.

Question 5. Raman: low-frequency Raman spectra in Fig. 2a should also be shown in a traditional way, as in the HP spectra in Fig. 2c. This way of presentation would better show strong broadening and central peak. The authors also wrote “the Raman modes for both inorganic cages (Fig. 2a) and organic cations (Fig. S1) in the $(BA_{0.73}PA_{0.27})_2PbI_4$ continuously broaden with increasing temperature, showing the absence of an abrupt change.” However, figure S1 shows spectra for $BA_{0.40}$, not $BA_{0.73}$. Furthermore, the temperature is given in Celsius degree instead of Kelvin, as in other figures and tables. Please correct this figure.

We thank the Reviewer for the careful reading. We have updated the mentioned figures, which are reproduced below.

Fig. 2. Dopant-induced pinning of the disordered phase under pressure and temperature tuning in $(BA)_2PbI_4$. Temperature-dependent (a) and pressure-dependent (b) low-frequency Raman

spectra of $(\text{BA})_2\text{PbI}_4$ and $(\text{BA}_{0.73}\text{PA}_{0.27})_2\text{PbI}_4$.

Fig. S2. Temperature-dependent Raman spectra for vibration modes of the organic cations of (a) $(\text{BA})_2\text{PbI}_4$, (b) $(\text{BA}_{0.73}\text{PA}_{0.27})_2\text{PbI}_4$, and (c) $(\text{PA})_2\text{PbI}_4$.

Question 6. Raman spectra: the authors noticed appearance of a new mode at 35 cm^{-1} for $(\text{BA}_{0.73}\text{PA}_{0.27})_2\text{PbI}_4$, not observed for the ordered BA_2PbI_4 . It seems also not present in the disordered BA_2PbI_4 . I do not understand origin of this new mode since if $(\text{BA}_{0.73}\text{PA}_{0.27})_2\text{PbI}_4$ and disordered BA_2PbI_4 have the same orthorhombic structure, their low-frequency spectra should be very similar. Can you propose origin of this mode, which is also clearly observed for $(\text{BA}_{0.18}\text{PA}_{0.82})_2\text{PbI}_4$ shown in figure S6? By the way, please show Raman spectra of the PA and PA0.82 samples in Fig. S6 in the same frequency range to facilitate comparison. I also do not see clearly additional splitting in PA_2PbI_4 compared to BA_2PbI_4 – please add Raman spectra presented in the traditional way to show the mentioned splitting.

We would like to note that the Raman spectra of doped structures indeed resemble the disordered phase of $(\text{BA})_2\text{PbI}_4$. By tracking the temperature-dependent evolution, we noted a pronounced redshift in this mode, shifting from 35 cm^{-1} to 28 cm^{-1} as temperature increased from 80 K to 300 K. This shift is accompanied by a significant broadening of the peak. Due to this significant redshift and broadening, the mode blends into an adjacent mode at 21 cm^{-1} , appearing as a peak shoulder (marked as * in Fig 2a) at high temperatures. Figure R1 illustrates the similarities between the Raman spectra of $(\text{BA})_2\text{PbI}_4$ and the doped samples. To clarify this,

we have revised the relevant sentences on page 6 of the manuscript.

Previous literature (J. Phys. Chem. C 2019, 123, 27904; ACS Nano. 2021, 15, 10153) has extensively explored the low-frequency vibrational modes of $(\text{BA})_2\text{PbI}_4$. Theoretical studies indicate that Raman frequencies below 40 cm^{-1} primarily involve the motion of the terminal iodide atoms (in the apical position) in the form of twisting of the octahedral axis. The 35 cm^{-1} mode, discerned through analysis of its polarization dependence, is attributed to the synergy of tilt and twist of PbI_6 octahedron. Since this mode is present in the disordered phase but absent in the ordered phase, we suggest it is related to the out-of-plane tilting of the octahedra, aligning with findings from previous studies. We have incorporated this discussion into page 6 of the revised manuscript.

Following the Reviewer's suggestion, we have modified the original Fig. S6 (now is Fig. S8) to better enable the comparison, as depicted below. In addition, we have added a new Fig. S9 to compare the Raman spectra of $(\text{BA})_2\text{PbI}_4$, $(\text{BA}_{0.18}\text{PA}_{0.82})_2\text{PbI}_4$, and $(\text{PA})_2\text{PbI}_4$. At 80 K, an additional peak splitting in $(\text{PA})_2\text{PbI}_4$ is observable in comparison to $(\text{BA})_2\text{PbI}_4$, particularly at the mode at around 26 cm^{-1} .

Fig. 2a. Temperature-dependent (a) low-frequency Raman spectra of $(\text{BA})_2\text{PbI}_4$ and $(\text{BA}_{0.73}\text{PA}_{0.27})_2\text{PbI}_4$. The temperature-dependent evolution of vibration mode of $(\text{BA}_{0.73}\text{PA}_{0.27})_2\text{PbI}_4$ at 35 cm^{-1} at 80 K is marked as dashed line and the peak shoulder is marked as *.

Fig. R1. Raman spectra of $(\text{BA})_2\text{PbI}_4$ and $(\text{BA}_{0.73}\text{PA}_{0.27})_2\text{PbI}_4$ at 300 K.

Fig. S8. Low-frequency temperature-dependent Raman spectra of (a) $(\text{BA})_2\text{PbI}_4$, (b) $(\text{BA}_{0.18}\text{PA}_{0.82})_2\text{PbI}_4$ and (c) $(\text{PA})_2\text{PbI}_4$.

Fig. S9. The comparison of low-frequency Raman spectra. (a) Temperature-dependent Raman spectra of $(\text{BA})_2\text{PbI}_4$ (top), $(\text{BA}_{0.18}\text{PA}_{0.82})_2\text{PbI}_4$ (middle) and $(\text{PA})_2\text{PbI}_4$ (bottom). (b) Raman spectra of $(\text{BA})_2\text{PbI}_4$, $(\text{BA}_{0.18}\text{PA}_{0.82})_2\text{PbI}_4$, and $(\text{PA})_2\text{PbI}_4$ at 80 K.

Question 7. The authors wrote “More disorder in the $(\text{BA}_{0.73}\text{PA}_{0.27})_2\text{PbI}_4$ can be seen by the larger atomic displacement parameters of the organic chain and iodide (Fig. S2).” I am not convinced that this conclusion is correct because the cif does not take into account presence of two different cations but only PA^+ (see my comment 3). Furthermore, Raman spectra of $(\text{BA}_{0.73}\text{PA}_{0.27})_2\text{PbI}_4$ presented in Fig. S2 show strong narrowing of bands on cooling and it seems that FWHM of the bands is slightly larger than FWHM of the corresponding bands of BA_2PbI_4 and PA_2PbI_4 . This suggests static rather than dynamic disorder at 80 K for the mixed BA-PA compounds. Numerous literature reports on lead halide perovskites also showed that organic cations are well-ordered at 80 K and, therefore, static disorder in the mixed-cation samples is a more reasonable explanation.

We appreciate the Reviewer for providing these insightful comments. We have removed the equivalent isotropic displacement parameters of organic cations from the original Fig. S2 (now is Fig. S3). Nevertheless, the same conclusion can be inferred from analyzing the inorganic atoms. We agree with the Reviewer’s assessment that static disorder is likely to play a more significant role in the doped structures as compared to $(\text{BA})_2\text{PbI}_4$ and $(\text{PA})_2\text{PbI}_4$ at 80

K. We have revised the corresponding text on page 7 of the revised manuscript.

Fig. S3. Equivalent isotropic displacement parameters (U_{eq}) of $(BA_{1-x}PA_x)_2PbI_4$ with different dopant ratios. Iodine in and out of the plane are denoted by I_1 and I_2 , respectively.

Question 8. Please check the text and figures for typos and errors. For instance, “decrease” in Fig. 4a. References should be corrected (subscripts, names added up to ten, abbreviation of journals etc.)

We have checked throughout the manuscript to correct the typos and errors. Thanks! We have formatted the reference to nature style, which include names only up to five.

Question 9. In the Structural Characterization and Acknowledgements information is given that NMR spectra were measured. However, no NMR spectra were presented and no information what they showed.

We have added the NMR spectra as Figure S13-17 in revised supplementary information.

Fig. S13 ^1H NMR spectrum (400 MHz, DMSO-d_6) of $(\text{BA})_2\text{PbI}_4$.

Fig. S14 ^1H NMR spectrum (400 MHz, DMSO- d_6) of $(\text{BA}_{0.73}\text{PA}_{0.27})_2\text{PbI}_4$.

Fig. S15 ^1H NMR spectrum (400 MHz, DMSO- d_6) of $(\text{BA}_{0.40}\text{PA}_{0.60})_2\text{PbI}_4$.

Fig. S16 ^1H NMR spectrum (400 MHz, DMSO- d_6) of $(\text{BA}_{0.18}\text{PA}_{0.82})_2\text{PbI}_4$.

Fig. S17 ^1H NMR spectrum (400 MHz, DMSO- d_6) of $(\text{PA})_2\text{PbI}_4$.

Reviewer: 3

Comments:

In this work, the authors have proven a novel and interesting method to create homojunction of different 2D perovskites. By cation doping they were able to tune the ordering transition of the spacer cation and the associated out-of-plane octahedral tilting in the inorganic framework, resulting in phase pinning of the disordered phase when decreasing temperature or increasing pressure. This is an innovative concept in material engineering for tuning perovskite crystalline order and thus modifying their optoelectronic properties at atomic scale. In addition, the authors have verified the exciton diffusion, which revealed interesting fundamental advance. I therefore propose to accept the work, after checking the following minor points:

We thank the Reviewer for the support of our work and valuable suggestions.

Question 1. what is the stability of the material under laser exposure for the measurements?

At low temperatures (e.g., 80 K), the samples remained stable even after 30 minutes of exposure to a 405 nm laser at 5 μ W (using a $\times 40$ objective, NA 0.60, Olympus), as shown in Fig. S18. The laser power was around 500 nW for the low-temperature PL measurements. Under ambient conditions, since the samples were less stable, we used a reduced laser power of 100 nW, which exhibited minimal laser-induced damage. For the Raman measurement, the samples were stable under 633 nm laser exposure at 10 mW. We have included the information in the Methods of the revised manuscript.

Fig. S18. PL spectra of $(\text{BA})_2\text{PbI}_4$ under a 5 μ W 405 nm laser exposure for 30 min at 80 K.

Question 2. What is the excitation density used?

The excitation power density was about 15 W cm^{-2} under ambient conditions and 15-50 W cm^{-2} at low-temperature and high-pressure conditions. We have included the information in the Methods of the revised manuscript

Question 3. Can the author retrieve the exciton binding energy?

We measured the absorption spectra of $(\text{BA}_{1-x}\text{PA}_x)_2\text{PbI}_4$, both disordered and ordered $(\text{BA})_2\text{PbI}_4$, and ordered $(\text{PA})_2\text{PbI}_4$ to determine their exciton binding energies (E_b), as depicted in Fig. S4. For these measurements, all samples were prepared as thin sheets by mechanical exfoliation. The exciton absorption energy (E_{ex}) was directly obtained from the peak position, while the band gap energy (E_g) was derived using the Tauc plot method. At room temperature, the doped structures $(\text{BA}_{1-x}\text{PA}_x)_2\text{PbI}_4$ and disordered $(\text{BA})_2\text{PbI}_4$ exhibit similar exciton binding energies ($E_b = E_g - E_{ex}$), approximately 0.38 eV, aligning with their isostructural phases. The value for $(\text{BA})_2\text{PbI}_4$ is consistent with the findings from previous studies (Nat. Commun. 2018,

9, 2254). In its ordered phase at room temperature, $(\text{PA})_2\text{PbI}_4$ exhibits an exciton binding energy of around 0.32 eV. The absorption spectrum of ordered $(\text{BA})_2\text{PbI}_4$ was also collected at 80 K, revealing an exciton binding energy of about 0.34 eV.

We have incorporated these new results on page 8 and Methods of the revised manuscript.

Fig. S4. Absorption spectra of various structures. Absorption spectra of (a) $(\text{BA})_2\text{PbI}_4$, (b) $(\text{BA}_{0.73}\text{PA}_{0.27})_2\text{PbI}_4$, (c) $(\text{BA}_{0.40}\text{PA}_{0.60})_2\text{PbI}_4$, (d) $(\text{BA}_{0.18}\text{PA}_{0.82})_2\text{PbI}_4$, and (e) $(\text{PA})_2\text{PbI}_4$ under ambient condition. (f) Absorption spectra of the ordered phase $(\text{BA})_2\text{PbI}_4$ at 80 K. Linear fitting using tauc plot method is represented by a red dashed line. The calculated exciton bonding energy are marked in the figure.

REVIEWERS' COMMENTS

Reviewer #1 (Remarks to the Author):

The authors revised the manuscript and all my concerns have been explained by adding additional data and revision of the text. I accept all the changes and the manuscript can be published in its present form. I congratulate the authors for very good results.

Reviewer #3 (Remarks to the Author):

The authors have provided compelling evidence and additional data which support the main conclusions. I propose to accept the work as is.